# Identification of Insertion/Deletion Markers for Photoperiod Sensitivity in Rice (*Oryza sativa* L.)

**DOI:** 10.3390/biology13050358

**Published:** 2024-05-20

**Authors:** Nguyen Thanh Tam, Dang Kieu Nhan

**Affiliations:** Mekong Delta Development Research Institute, Can Tho University, Campus 2, 3-2 Street, Can Tho 94115, Vietnam

**Keywords:** indica rice, INDEL, Mekong Delta, photoperiod sensitivity, sequencing

## Abstract

**Simple Summary:**

Photoperiod sensitivity is important for rice breeders to enable crop adaptation to changing climatic conditions. This study aims to pinpoint candidate insertion/deletion (INDEL) markers linked to photoperiod sensitivity in indica rice cultivars. The INDEL marker can be utilized to explore the gene function in rice flowering, potentially facilitating gene editing using CRISPR/Cas9 technology. Applying this INDEL marker in breeding programs can enhance the genetic gain and economic efficiency of a targeted breeding strategy.

**Abstract:**

The current study aims to identify candidate insertion/deletion (INDEL) markers associated with photoperiod sensitivity (PS) in rice landraces from the Vietnamese Mekong Delta. The whole-genome sequencing of 20 accessions was conducted to analyze INDEL variations between two photoperiod-sensitivity groups. A total of 2240 INDELs were identified between the two photoperiod-sensitivity groups. The selection criteria included INDELs with insertions or deletions of at least 20 base pairs within the improved rice group. Six INDELs were discovered on chromosomes 01 (5 INDELs) and 6 (1 INDEL), and two genes were identified: *LOC_Os01g23780* and *LOC_Os01g36500*. The gene *LOC_Os01g23780*, which may be involved in rice flowering, was identified in a 20 bp deletion on chromosome 01 from the improved rice accession group. A marker was devised for this gene, indicating a polymorphism rate of 20%. Remarkably, 20% of the materials comprised improved rice accessions. This INDEL marker could explain 100% of the observed distinctions. Further analysis of the mapping population demonstrated that an INDEL marker associated with the *MADS-box* gene on chromosome 01 was linked to photoperiod sensitivity. The F1 population displayed two bands across all hybrid individuals. The marker demonstrates efficacy in distinguishing improved rice accessions within the indica accessions. This study underscores the potential applicability of the INDEL marker in breeding strategies.

## 1. Introduction

Rice (*Oryza sativa*) production is important in the global food supply, particularly in Asian countries. Rice is grown in diverse climatic regions with wide variation in their photoperiods [1,2]. It is inherently a short-day plant, displaying pronounced photoperiod sensitivity, where short days encourage flowering while long days inhibit it [3]. The heading date (flowering time) is an important trait for the regional and seasonal adaptation of rice and influences grain yield [4]. Thus, photoperiod-sensitive control in rice is essential for the production of rice adaptive to different climatic conditions [5].

The primary distinction between landrace and improved rice varieties lies in their photoperiod sensitivity [5]. Rice landraces flower under short-day conditions [5], while improved rice can flower throughout the year, given sufficient time for the preproduction stage. Previous studies on heading dates focused on genotypic photoperiod-sensitive rice cultivars [6]. The heading date is regulated by multiple quantitative trait loci (QTLs) [7] with several well-known QTLs identified to date, including *se-1* [8,9], *Hd1* [10], *Hd6* [11], *Hd3a* [12], *Ehd1* [13], *Ghd7* [14], *DTH8* [15], *Ghd8* [16], *dth3* [17], *Hd17* [18], *Ehd4* [19], *Hd16* [20], *RFT1* [21], *DTH2* [22], and *DTH7* [23]. Furthermore, the intricate regulation of strong photoperiod sensitivity involves collaborative and competitive interactions among *Hd1*, *Ghd7*, and *DTH8* in rice heading [3]. Consequently, the heading date exhibits a strong correlation with photoperiod sensitivity. While previous studies focused on identifying QTLs using marker-assisted selection and backcrossing, recent years have seen the adoption of genome-wide association studies (GWASs) as a method for identifying candidate genes associated with phenotypic traits, including the heading date [24].

The application of single-nucleotide polymorphism (SNP) genotyping methods based on next-generation sequencing and high-throughput genotyping has gained popularity in various crops [25,26,27,28,29]. GWAS, facilitated by SNP analysis, enables the study of complex quantitative traits such as plant stress tolerance, growth, and development.

Insertion/deletion (INDEL) markers are valuable markers in rice breeding [30], aiding in distinguishing the phenotypic traits of rice varieties [31,32,33,34,35]. However, existing studies have primarily utilized available INDEL markers from the Nipponbare genomic reference database. The distinction between photoperiod-sensitive and non-sensitive rice based on whole-genome sequencing and INDEL markers has not been a primary focus of previous studies. In this study, we employed the whole-genome sequencing of 20 indica rice accessions (including 16 landraces and four improved rice accessions) collected from the Vietnamese Mekong Delta provinces to analyze the relationship between two distinct groups based on INDEL information.

## 2. Methods

### 2.1. Plant Materials

A total of twenty rice accessions from a pool of 99 rice accessions were selected following the criteria outlined by Tam et al. [36]. The selected accessions comprised sixteen landraces and four improved rice accessions originating from the Vietnamese Mekong Delta provinces (Table 1). The origin locations of the studied rice landraces are characterised by salinity-affected and rainfed agro-ecology. In contrast, the origins of the improved rice accessions are considered freshwater and irrigated agro-ecology. 

### 2.2. Genome Sequencing and Data Analysis

#### 2.2.1. Genomic DNA Isolation

Genomic DNA was extracted using the DNeasy Plant Mini kit (Qiagen, Hilden, Germany), adhering to the manufacturer’s protocol. Sample preparation of leaf specimens and DNA quality assessment procedures were followed [36]. Sterilized seeds of each accession were placed in Petri dishes for germination. Three days after germination, Petri dishes were transferred into a growth chamber at 27 °C. In a period of 7 to 10 days after germination, leaves were harvested for DNA extraction. The quantity of DNA was measured using a Thermo Scientific NanoDrop 2000 spectrophotometer (Fisher Scientific, Hampton, NH, USA) with a volume of 1 µL. DNA quality of samples was checked by 1% (*w*/*v*) agarose gel electrophoresis. 

#### 2.2.2. Whole-Genome Sequencing

The whole genome of the selected varieties was sequenced using Illumina’s next-generation sequencing approach. Library preparation and sequencing were performed at each step of the procedure. The workflow was as follows: (1) checking of DNA; (2) construction of libraries; (3) checking of libraries; (4) sequencing.

We performed two main methods of quality control for DNA samples: (1) agarose gel electrophoresis to test DNA degradation and potential contamination, and (2) Qubit 2.0 to quantify DNA concentration precisely.

Library construction and quality control: A total amount of 1.0 μg DNA per sample was used as input material for DNA sample preparation. Sequencing libraries were generated using NEBNextR DNA Library Prep Kit (Novogene corporation INC (823 Anchorage Place, Chula Vista, CA, USA)) following manufacturer’s recommendations and indices were added to each sample. Genomic DNA was randomly fragmented to a size of 350 bp by shearing; then, DNA fragments were end-polished, A-tailed, and ligated with an NEBNext adapter for Illmina sequencing, and further PCR-enriched by P5 and indexed P7 oligos. PCR products were purified (AMPure XP system), and resulting libraries were analyzed for size distribution by an Agilent 2100 Bioanalyzer (Santa Clara, CA, USA) and quantified using real-time PCR.

Sequencing: Qualified libraries were fed into Illumina sequencers after pooling according to their effective concentration and expected data volume.

#### 2.2.3. Data Processing

Processing of INDELs for the twenty varieties followed the protocol outlined by Tam et al. [36]. Sequenced raw reads underwent filtering and sorting based on original sample names. Sequence trimming was executed to a length of 350 bp using a Trimmomatic with the following specified parameters: LEADING:19, TRAILING:19, SLIDINGWINDOW:30:20, AVGQUAL:20, and MINLEN:51. High-quality reads were mapped to the Nipponbare IRGSP1.0 reference genome using Bowtie2 [37], available in Galaxy (www.https://usegalaxy.org, accessed on 24 November 2023). Further filtering was performed using Picard in Galaxy, and alignments were adjusted around INDELs using the INDELRealigner tool in Genome Analysis Toolkit v2.8 [38]. INDEL calling utilized the Unified Genotyper tool in GATK v2.8 [39], and the initial INDEL dataset was filtered based on the following parameters: missing call rate (MCR), 100%; minor allele frequency (MAF), 0.05; and heterozygosity rate (HR), 0.0. Subsequent dataset filtering included functions to separate accessions into two groups (landrace and improved), leading to the identification of 2240 INDELs for further analysis.

### 2.3. Identifying INDELs between Photoperiod-Sensitive and Non-Sensitive Rice Groups

The INDELs, derived from dataset filtering (2240 INDELs) between the two groups, with a minimum length of 20 bp, were specifically chosen to emphasize the distinct variations between landrace and improved rice. In a previous study, Adedze et al. selected INDEL sizes starting from 30 bp as agarose gel PCR-based markers [40]. The selected INDELs (candidates) were employed for gene filtering using the Rice SNP-Seek Database (https://snp-seek.irri.org/, accessed on 7 December 2023). Candidate INDEL positions were utilized to search the Rice SNP-Seek Database using the Search/Genotype function.

### 2.4. Primer Design for Testing Improved Rice Varieties

Primer design, confirming the differences between landrace and improved groups, was utilized on the INDEL located on the gene in the Rice SNP-Seek Database. The primer, named Photoperiod-sensitivity (PS), was designed using Primer3Plus (http://primer3plus.ut.ee/cgi-bin/primer3plus/primer3plus.cgi, accessed on 7 December 2023). The information began with the specified primers (5′-GAGGGAGCTCTCCATCCTCT-3′ for the left primer and 5′GCTTCAACTCGAGGCACTCT-3′ for the right primer), targeting the region on chromosome 01. This amplification product was 203 from the standard sequence referenced for the Nipponbare variety.

### 2.5. Validation of INDEL Candidate from the Rice SNP-Seek Database

The validation of potential INDEL candidates based on the Rice SNP-Seek Database (3K) is a pivotal step. This database is well known for SNP, INDEL, gene, and other genomic data from 3024 whole-genome sequencing accessions [41]. Mansueto et al. categorized these accessions into twelve distinct sub-populations based on SNP and INDEL profiles, including admix, aro, aus, Ind1a, ind1b, ind2, ind3, indx, japx, subtrop, temp, and trop [41]. In order to ascertain the credibility of INDEL candidates in our study, an investigation was conducted within the rice genome utilizing the 3K database. This involved the genomic positions of the INDEL candidates across the 3K dataset. The diversity of INDEL markers was assessed, focusing on the occurrence of Nipponbare reference genome sequences or deletions within each sub-population.

### 2.6. Data Analysis

The filtering process utilized TASSEL 5.2.50 (Trait Analysis by Association Evolution and Linkage) [42]. Candidate genes were scrutinized using the QTL and gene database in the Rice SNP-Seek Database [41]. The variations among the twelve rice groups in the Rice SNP-Seek Database, and the ratio of deletion accessions to total accessions was calculated for each group.

## 3. Results

### 3.1. INDEL Information on Selected Landraces and Improved Accessions

A total of 9690 INDELs were identified through the whole-genome sequencing of 20 accessions. The highest count of INDELs was observed on Chr. 01 (2970 INDELs), followed by Chr. 06 (1300 INDELs), while the lowest count was on Chr. 09, with 241 INDELs (Table 2). Applying filtering criteria (MCR: 100%, MAF: 0.0, max allele frequency: 0.0), the total INDELs in the landrace group (2240) were fewer than those in the improved group (5740) (Table 2). After combining and further filtering with MCR (100%), MAF (0.2), max allele frequency (1.0), and HR (0.0), 2240 INDELs were identified, with a noteworthy location on Chr. 01 (1623 out of 2240 INDELs). No differences were found between the two groups in Chr. 09 and Chr. 10. Chr. 02, Chr. 05, Chr. 11, and Chr. 12 exhibited only a few INDELs (Table 2).

This study focused on insertions or deletions between the two groups, with a minimum length of 20 bp. This analysis revealed 22 INDEL markers, of which 14 were on Chr. 01, 1 was on Chr. 03, 2 were on Chr. 04, 2 were on Chr. 06, and 3 were on Chr. 07 (Table 3). Additionally, only one INDEL marker (S01_14359423) involved an insertion; the rest were deletions. This implies that 21 INDEL markers differed from the Nipponbare genome (Japonica rice cultivar). Almost all genome types of the improved group differed from the genome reference, except for three INDEL markers (S01_14359423, S06_24284080, and S07_1972486) (Table 3).

### 3.2. Gene Related to Photoperiod Sensitivity

To identify candidate genes that are associated with photoperiod sensitivity, the locations of the 22 INDELs were examined on the website https://snp-seek.irri.org/ (accessed 10 June 2023). Ten INDELs located in 10 genes were found in landrace and improved rice accessions. Each gene had one INDEL. Seven genes were located on chromosome 01, two on chromosome 04, and one on chromosome 07 (Table 4). The functions of these genes included *MADS-box* family genes, acyl-ACP thioesterase, DUF26 kinases, carboxyl-terminal peptidase, pentatricopeptide, CBS domain-containing protein, homeobox-associated leucine zipper, WD domain, G-beta repeat domain-containing protein, and CAMK_KIN1/SNF1/Nim1_like_AMPKh.1—CAMK, the latter of which includes calcium/calmodulin-dependent protein kinases. One gene (*LOC_Os01g23780*), encoding a putative regulator of rice flowering [43], was identified from 13373620 to 13374665 on chromosome 01. In this gene’s position, the improved rice accession group exhibited a 20 bp deletion compared to the landrace accession group.

### 3.3. Differences in Photoperiod-Sensitivity Group in Gene LOC_Os01g23780

The MADS-box family gene *LOC_Os01g23780*, which may be involved in rice flowering [43], was located in the INDEL region between the landrace (photoperiod sensitive) and improved varieties (photoperiod insensitive). We therefore designed the primer based on Primer3Plus (http://primer3plus.ut.ee/cgi-bin/primer3plus/primer3plus.cgi, accessed on 7 December 2023), and we obtained the photoperiod-sensitive primer (PS primer) with the following information: left primer 5′-GAGGGAGCTCTCCATCCTCT-3′ and right primer 5′GCTTCAACTCGAGGCACTCT-3′). It occurred at positions 13373620 to 13373910 in chromosome 01, and produced two prominent diagnostic fragments of approximately 203 bp and 183 bp for landrace rice and improved rice, respectively. In this position, the improved rice accession group was deleted at 20bp from the normal sequence of Nipponbare references or indica landrace rice.

To confirm the phenotype diversity of two distinct photoperiod-sensitive groups, we designed a primer to screen the action of the gene *LOC_Os01g23780*. The results indicate that the PCR product of the landrace accession group was larger than that of the improved group at about 20bp (Figure 1). Each of the 16 landrace accessions exhibited a PCR product measuring 203 bp. In contrast, the four improved rice accessions displayed a band of 183 bp (Figure 1). In addition, Figure 2 illustrates that the F1 population resulting from the cross between Nang Thom Cho Dao and MTL372 exhibited two bands across all hybrid individuals. Specifically, the Nang Thom Cho Dao accession displayed a band of 203 bp, while MTL372 exhibited a band of 183 bp.

Phenotypic analyses of heading dates and agronomic traits were carried out on 20 accessions in Hong Dan district, Bac Lieu province (latitude: 9°29′20.6″ N; longitude: 105°30′24.1″ E) from September 2020 to March 2021. Among these accessions, the heading dates of 16 landraces ranged from 24 December 2020, to 31 January 2021 (Appendix A). In contrast, four improved accessions were not affected by short-day conditions, displaying heading patterns determined by their genetic traits (Appendix A). The improved-accession group displayed shorter plant heights in comparison to the landrace group (Appendix A). 

The INDEL marker S01_13373751, located in the gene *LOC_Os01g23780*, was employed to investigate the genetic diversity within the rice genome via the Rice SNP-Seek database (3K). It revealed that five sub-populations (aro, japx, subtrop, temp, and trop) exhibited identical genotypes to those of the Nipponbare and landrace groups (Table 5). Therefore, this INDEL marker distinguishes landrace and improved groups and separates the genotype of the five subgroups above from the improved indica rice group (Table 5).

## 4. Discussion

Culturing improved rice varieties with photoperiod insensitivity and a short growth duration has been prominent in Vietnam and other Asian countries, driven by agricultural intensification for enhanced food security and export [44]. Photoperiod sensitivity is important to rice breeders [45]. In the Vietnamese Mekong Delta, since the 1980s, improved rice varieties have gradually replaced rice landraces with photoperiod sensitivity and long growth durations [46]. The key distinction between landrace and improved rice lies in their responsiveness to day length, impacting crucial aspects such as flowering, grain yield, and overall rice productivity. To investigate photoperiod sensitivity, numerous earlier studies have concentrated on the heading date [3,6,45] because rice typically utilizes photoperiodism to regulate the timing of flowering. The progression of rice development leading up to floral initiation is characterized by two consecutive phases: the basic vegetative growth phase, which is photoperiod-insensitive, and the photoperiod-sensitive phase [47].

INDEL polymorphisms are the second most prevalent type of genetic variation, following SNPs, in humans and plants [48]. In addition, INDEL markers easily distinguish the genotypes among individuals based on their size. Therefore, developing INDEL markers is becoming popular for crop genetic studies [48]. 

In our study, identifying 2240 INDELs from the genome sequencing of 20 accessions, representing two distinct groups of landrace and improved rice, revealed significant differences. Chr. 01 exhibited the highest discrepancy, with 1263 INDELs, followed by chr. 04 (220 INDELs), 06 (171 INDELs), and 03 (123 INDELs), collectively accounting for 79.3% of all INDELs (1777 INDELs). The information on these INDELs can be found in the 3K database, where the highest number of INDELs is in chr. 01 [41]. Fourteen INDELs in Chr. 01 exhibited insertions or deletions of at least 20 bp, suggesting their potential utility for distinguishing between the two groups. An INDEL at the position of 13373751 bp on chr. 01, involving a 20 bp deletion in the improved rice group compared to the landrace group, was identified. This INDEL resides in the codon of the gene *LOC_Os01g23780*, classified as *OsMADS95—MADS-box*, a gene known for its involvement in floral organ development, flower development, and overall plant growth and development [43]. *LOC_Os01g23780* plays a crucial role during the reproduction phase of rice [43].

The INDEL candidate marker for the gene *LOC_Os01g23780* successfully distinguished landrace and improved rice groups. Utilizing this INDEL marker allows breeders to assess the outputs of crosses between landrace and improved rice parents. Successful crossings can be identified by the presence of two bands in the F1 individual using the PS marker. Additionally, applying this marker can enhance the genetic gain and economic efficiency of a targeted breeding strategy [49]. Furthermore, this INDEL marker exhibited broader applicability, enabling the classification of five subgroups (aro, japx, subtrop, temp, and trop) within the indica improved rice category (Table 5). This finding underscores the potential significance of the identified INDEL marker in both distinguishing and categorizing rice accessions, providing valuable insights for further breeding programs and genetic studies.

## 5. Conclusions

This study was undertaken to elucidate the genetic distinctions, specifically the number of INDELs, between landrace and improved rice groups. A significant discovery is that a substantial proportion of the identified INDELs, totaling almost 2240, were concentrated on chromosome 01. In addition, the specific INDEL position S01_13373751 bp within the gene *LOC_Os01g23780* emerged as a focal point of significance.

The outcomes of the current study lend strong support to the proposition that the INDEL marker S01_13373751 holds practical utility. This marker demonstrates efficacy in distinguishing improved rice accessions within the indica accessions. Further, it serves as a discriminative tool between improved rice accessions originating from indica and Japonica rice accessions. This study underscores the potential applicability of the INDEL marker in breeding strategies.

## Figures and Tables

**Figure 1 biology-13-00358-f001:**
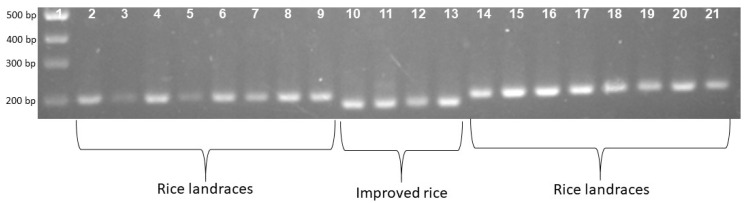
Gel electrophoresis patterns depicting locus phase amplification across all 20 genotypes. Rice landrace produces a PCR product with a size of 203 bp. Improved rice produces a PCR product with a size of 183 bp. 1: Marker labels; 2–9: rice landraces; 10–13: improved rice; 14–21: rice landraces.

**Figure 2 biology-13-00358-f002:**
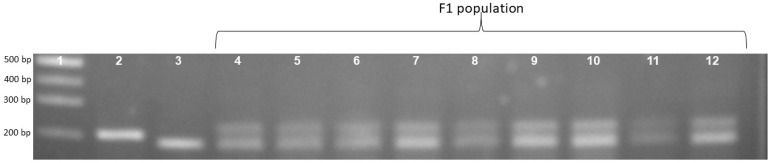
Gel electrophoresis image illustrating locus phase amplification patterns in the F1 population. Rice landrace produces a PCR product with a size of 203 bp and improved rice produces a PCR product with a size of 183 bp, whereas F1 population has bands with sizes of 203 bp and 183 bp for whole hybrid population. 1: Marker labels; 2: Nang Thom Cho Dao (rice landraces); 3: MTL372 (Improved rice); 4–12: F1.

**Table 1 biology-13-00358-t001:** Rice accession codes and names, origin provinces, and photoperiod sensitivity.

No.	Accession Codes	Accession Names	Provinces	Sensitivity
1	MDI-1	BO LIEP 2	Ca Mau	Sensitive
2	MDI-7	LUN VANG	Ca Mau	Sensitive
3	MDI-18	NEP SUA	Ca Mau	Sensitive
4	MDI-21	DOC PHUNG	Ben Tre	Sensitive
5	MDI-22	TRA LONG 2	Ca Mau	Sensitive
6	MDI-31	MOT BUI DO CAO CA MAU	Ca Mau	Sensitive
7	MDI-41	THOM MUA 1	Ca Mau	Sensitive
8	MDI-42	HUYET RONG LONG AN	Long An	Sensitive
9	MDI-44	MTL110	Ben Tre	Insensitive
10	MDI-61	MTL938	Can Tho	Insensitive
11	MDI-64	MTL930	Can Tho	Insensitive
12	MDI-66	MTL372	Can Tho	Insensitive
13	MDI-72	CHAU HANG VO	Kien Giang	Sensitive
14	MDI-88	NANG KEO BA TU	Tra Vinh	Sensitive
15	MDI-96	NEP MO 3	Ca Mau	Sensitive
16	MDI-102	SOI DO	Tra Vinh	Sensitive
17	MDI-123	NEP AO GIA	Kien Giang	Sensitive
18	MDI-125	NHO HUONG	Ben Tre	Sensitive
19	MDI-126	NANG THOM CHO DAO	Long An	Sensitive
20	MDI-133	CHO BIEN	Kien Giang	Sensitive

**Table 2 biology-13-00358-t002:** Number of INDELs identified in rice accessions based on 20 whole-genome sequences.

Chr.	20 Accessions	Landraces	Improved Varieties
1	2970	1623	2559
2	641	4	193
3	978	123	658
4	604	220	434
5	311	6	126
6	1300	171	748
7	289	43	164
8	659	34	261
9	241	0	55
10	600	0	248
11	528	9	178
12	569	7	116
Total	9690	2240	5740

**Table 3 biology-13-00358-t003:** Different characteristics of INDELs with minimum lengths of 20 bp between two accession groups.

INDEL Candidate	Alleles	Reference	Alternative	Chr.	Position	Alleles of Landrace Group	Alleles of Improved Group	Number of Alleles from Reference	Number of Alleles from Alternative
S01_13373751	G/-	AGACGGCGGCGGTGCGGTGTC	A	1	13,373,751	G	-	21	1
S01_14359423	-/G	A	AGATCGCATCGTCTTCTTTCCCG	1	14,359,423	-	G	1	23
S01_14804579	G/-	AGAGCCACGCGTCCGAATCCGGGT	A	1	14,804,579	G	-	24	1
S01_17388348	A/-	AAACTCTTTAACTTTTTTAAGT	A	1	17,388,348	A	-	22	1
S01_17952375	A/-	AATTTGAGTTGAAAATTTTCAAT	A	1	17,952,375	A	-	23	1
S01_18759075	G/-	AGGTTAATTTTTTATGGGAC	A	1	18,759,075	G	-	20	1
S01_19437208	C/-	CCCGCCGCCGGGCCATTGTCGCCACCT	C	1	19,437,208	C	-	27	1
S01_20258054	C/-	TCTTAGGCAAATAGTAAATTCTCCC	T	1	20,258,054	C	-	25	1
S01_20267277	C/-	TCCCTTTGATTTTCAACATTTG	T	1	20,267,277	C	-	22	1
S01_20340904	T/-	ATTGAAAATAAATGAAACTGGTT	A	1	20,340,904	T	-	23	1
S01_20539068	T/-	ATCTAACCTATATTTATAGAG	A	1	20,539,068	T	-	21	1
S01_42275186	C/-	GCAAATGTTTACTATAGCAC	G	1	42,275,186	C	-	20	1
S01_42299024	C/-	ACTTTCTTCTCATATGTCACTTC	A	1	42,299,024	C	-	23	1
S01_42380745	A/-	TACATAAGCTTTCAACATTTTCCTTCA	T	1	42,380,745	A	-	27	1
S03_26306533	A/-	AAAAACTCAAAGCGGCAGGTATTTTAT	A	3	26,306,533	A	-	27	1
S04_27748597	T/-	CTATAAACATATTTTAAAGA	C	4	27,748,597	T	-	20	1
S04_27795695	T/-	ATCTGTCATCTCATCTTTAAAGT	A	4	27,795,695	T	-	23	1
S06_19237696	A/-	GAGATCCATGCTTGGCACCCTTC	G	6	19,237,696	A	-	23	1
S06_24284080	-/C	CCTGTCCATCTCACCGGAGATGCTCT	C	6	24,284,080	-	C	26	1
S07_1972486	-/G	TGGACTAAAGTTTTACTTTAGG	T	7	1,972,486	-	G	22	1
S07_26941649	T/-	GTAGTGATAAAATGAGAACC	G	7	26,941,649	T	-	20	1
S07_5102539	G/-	GGGCGGCGGCGGCGCGGCGTC	G	7	5,102,539	G	-	21	1

(-): deletion alleles from the reference genome.

**Table 4 biology-13-00358-t004:** Comprehensive INDEL information for the 20 selected accessions.

INDEL Markers	Alleles	Gene Names	Gene Positions	Functions
S01_13373751	G/-	LOC_Os01g23780	chr01:13373620..13374665 (+ strand)	OsMADS95—MADS-box family gene with M-beta type-box, expressed
S01_14359423	-/G	-	-	-
S01_14804579	G/-	-	-	-
S01_17388348	A/-	LOC_Os01g31760	chr01:17384240..17389590 (+ strand)	Acyl-ACP thioesterase, putative, expressed
S01_17952375	A/-	-	-	-
S01_18759075	G/-	-	-	-
S01_19437208	C/-	LOC_Os01g35110	chr01:19436943..19437437 (- strand)	Expressed protein
S01_20258054	C/-	LOC_Os01g36500	chr01:20256262..20260718 (- strand)	TKL_IRAK_DUF26-lh.5—DUF26 kinases have homology to DUF26-containing loci, expressed
S01_20267277	C/-	-	-	-
S01_20340904	T/-	LOC_Os01g36660	chr01:20338108..20341899 (- strand)	Carboxyl-terminal peptidase, putative, expressed
S01_20539068	T/-	-	-	-
S01_42275186	C/-	-	-	-
S01_42299024	C/-	LOC_Os01g72930	chr01:42297255..42300253 (+ strand)	Pentatricopeptide, putative, expressed
S01_42380745	A/-	LOC_Os01g73040	chr01:42377194..42381600 (+ strand)	CBS domain-containing protein, putative, expressed
S03_26306533	A/-	-	-	-
S04_27748597	T/-	LOC_Os04g46350	chr04:27477579..27479717 (- strand)	Homeobox-associated leucine zipper, putative, expressed
S04_27795695	T/-	LOC_Os04g46892	chr04:27795138..27798763 (- strand)	WD domain, G-beta repeat domain-containing protein, expressed
S06_19237696	A/-	-	-	-
S06_24284080	-/C	-	-	-
S07_1972486	-/G	-	-	-
S07_26941649	T/-	-	-	-
S07_5102539	G/-	LOC_Os07g09610	chr07:5102509..5107298 (+ strand)	CAMK_KIN1/SNF1/Nim1_like_AMPKh.1—CAMK includes calcium/calmodulin-dependent protein kinases, expressed

(-): no genes consist of INDEL candidate.

**Table 5 biology-13-00358-t005:** Number of accessions displaying deletion at position 13373751 on chromosome 01 from the 3K database.

Sub-Populations	Number of Accessions	Deletion Accessions	
		Number	%
Ind1A	209	18	8.6
Ind1B	205	16	7.8
Ind2	285	42	14.7
Ind3	475	34	7.2
Indx	615	51	8.3
admix	103	4	3.9
aus	201	54	26.9
aro	76	0	0
japx	83	0	0
subtrop	112	0	0
temp	288	0	0
trop	372	0	0
Total	3024	219	7.2

Rice accessions are from the 3K rice genome dataset. The 3K rice genome accessions consist of 209 Ind1A, 205 Ind1B, 285 Ind2, 475 Ind3, 615 Indx, 103 admix, 201 aus, 76 aro, 83 japx, 112 subtrop, 288 temp, and 372 trop accessions.

## Data Availability

The sequence data of the 20-accession MDI set generated in this study have been deposited in the NCBI database (accession number PRJNA787403).

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
