# Peer review of "Identification of Insertion/Deletion Markers for Photoperiod Sensitivity in Rice (Oryza sativa L.)"

_biology, 2024, doi:10.3390/biology13050358_

Round 1
Reviewer 1 Report
Comments and Suggestions for Authors
Major remarks
- The introduction section of the manuscript does not provide clue on its rational and potential benefit, given the fact that the issue it is dealing with (genetics of rice photosensitivity and flowering time) has already given rise to hundreds of publications and gene cloning works.
- Materiel & Methods:
o The rational of resorting to Indel markers, and among these markers selecting the ones with the minimum length of 20 is not clear.
o The approach implemented for candidate gene detection (loking genotypic contrast between two populations of small and unequal sizes (16 & 4 individuals) has not the appropriate detection power and does not allow accounting for confounding effects, while the two populations differ for a large number of phenotypic traits and have low intra-population phenotypic variability.
o The whole M&M section is presented too approximatively to allow for a thorough review.
- Results:
- The term “significant” is often used while no statistical analysis was performed.
- As consequence of the inadequate methodological approach, there is a mix-up between correlation and causal relation.
- The rational of the choice of the candidate gene LOC_Os01g23780 for further "confirmation" is not clear.
- The so called “high correlation between” the “Deletion at Position 13373751” and photosensitivity is contradicted by the presence of this deletion in almost all component of the indica group that includ both landraces and improved accessions.
Discussion : the Findings of the work is not confronted to literature on the genetics genes network involved in rice photosensitivity. The literature cited is relatd to heading date in rice not on photosensitivity.
Minor remarks: The manuscript is crippled with sentences that need reformulation (for lack of clarity) or precision (for vagueness). Plese see my annotation on the manuscript.

Author Response
Comment 1: The introduction section of the manuscript does not provide clue on its rational and potential benefit, given the fact that the issue it is dealing with (genetics of rice photosensitivity and flowering time) has already given rise to hundreds of publications and gene cloning works. Answer: We have augmented the introduction with further elaboration on the rationale and potential benefits of understanding the relationship between heading date, flowering time, and photoperiod sensitivity.
Comment 2: The rational of resorting to Indel markers, and among these markers selecting the ones with the minimum length of 20 is not clear. Answer: In this investigation, we conducted a comprehensive filtering of all Indels in the dataset. However, it is noteworthy to highlight specific candidate indels that hold potential for marker development. We posit that a gap of approximately 20 base pairs between photoperiod-sensitive and insensitive variants facilitates a straightforward differentiation between them.
Comment 3: The approach implemented for candidate gene detection (looking genotypic contrast between two populations of small and unequal sizes (16 & 4 individuals) has not the appropriate detection power and does not allow accounting for confounding effects, while the two populations differ for a large number of phenotypic traits and have low intra-population phenotypic variability. Answer: Considering the limited size of the population, we employed the candidate indel marker to identify variations within the 3K database (Rice SNP-Seek Database).
Comment 4: The term “significant” is often used while no statistical analysis was performed. Answer: We already deleted in the manuscript.
Comment 5: The rational of the choice of the candidate gene LOC_Os01g23780 for further "confirmation" is not clear. Answer: Because the gene function of LOC_Os01g23780 is as OsMADS95 - MADS-box, a gene known for its involvement in floral organ development, flower development, and overall plant growth and development (Arora et al., 2007). Moreover, LOC_Os01g23780 plays a crucial role during the reproduction phase of rice (Arora et al., 2007). This is explain in the discussion section of the manuscript.
Comment 6: The so called “high correlation between” the “Deletion at Position 13373751” and photosensitivity is contradicted by the presence of this deletion in almost all component of the indica group that include both landraces and improved accessions.
Answer: In this study, we conducted a comparison between landraces and improved accessions, utilizing the Nipponbare sequence as a reference. Consequently, our indel analysis also involved a comparison with the Nipponbare cultivar, with a specific focus on positions that exhibit differences between the landrace and improved groups.
Comment 7: Discussion : the Findings of the work is not confronted to literature on the genetics genes network involved in rice photosensitivity. The literature cited is related to heading date in rice not on photosensitivity. Answer: We have augmented the discussion with further elaboration on photoperiod sensitivity because rice typically utilize photoperiodism to regulate the timing of flowering. The progression of rice development leading up to floral initiation is characterized by two consecutive phases: the basic vegetative growth phase, which is photoperiod-insensitive, and the photoperiod-sensitive phase.
Comment 8: Minor remarks: The manuscript is crippled with sentences that need reformulation (for lack of clarity) or precision (for vagueness). Please see my annotation on the manuscript. Answer: Thank you very much, we changed the information on the manuscript based on your suggestions.
Reviewer 2 Report
Comments and Suggestions for Authors
The authors analyzed SNPs in 20 indica rice accessions from Mekong Delta regions in Vietnam and found the insertion/deletion (Indel) marker that can discriminate local and improved varieties. In addition, they discussed the association with the phenotypic variance. The topic is important from both academic and practical aspects.
However, the authors should consider the following points.
Major points:
Association between SNPs and photosensitivity
Their recent work (Tam et al., 2019, doi:10.3835/plantgenome2019.06.0042) revealed the diversity of genome sequence in a large number of indica rice varieties including Mekong Delta rice landraces. The present study further developed the PCR-based Indel marker that can be applied at least into 20 indica rice accessions.
However, to discuss the association with the phenotypic variance (i.e., photosensitivity), a more in-depth analysis such as GWAS using large numbers of varieties have been required. In contrast, because the authors investigated the indels in a small number (20) of varieties, it is difficult to conclude the association with photosensitivity. They should analyze a larger number of rice varieties to discuss the association with photosensitivity. At least, the authors should discuss the limitations of their study.
In addition, the meaning of “high correlation (of the gene LOC_Os01g23780) with photosensitivity” (P7L168) is not clear. Which data and/or statistical analysis indicate the “high correlation”? The author should explain how the “high correlation” was detected more in details.
From a plant physiological aspect, it may be interesting that the gene LOC_Os01g23780 that would be involved in floral development in rice (Arora et al. 2007) was identified as a putative responsible regulator of photosensitivity. The authors may describe a requirement of further investigation about how the Indel in the gene LOC_Os01g23780 can alter rice photosensitivity.
Minor points:
Introduction
P2L41-42: “several well-known QTLs” → In fact, many (more than “several”) QTLs were described below (P2L42-47). The description “several well-known QTLs…” should be revised.
P2L49: “Genome-wide association study (GWAS)” → genome-wide association study (GWAS)
P2L52: “Single Nucleotide Polymorphism (SNP)” → single nucleotide polymorphism (SNP)
Method
P4L114-115: “the https://snp-seek.irri.org/ database” → the Rice SNP-Seek Database (https://snp-seek.irri.org/)
Results
P5L143: “focused focused” → focused
P9L176: “photoperiod-sensithive” → photoperiod-sensitive
P10L201: “genotypr” → “genotype”
Discussion
P11L226-233: A similar description about QTLs identified previously was found also in the introduction part (P2L42-47). The repetition should be avoided.
Tables 1-5 and Figures 1-2:
More detailed descriptions about tables and figures, such as statistical analysis, data source, DNA size marker (used in Figures 1 and 2) and abbreviation, etc. should be described.
Author Response
Comment 1: Association between SNPs and photosensitivity Their recent work (Tam et al., 2019, doi:10.3835/plantgenome2019.06.0042) revealed the diversity of genome sequence in a large number of indica rice varieties including Mekong Delta rice landraces. The present study further developed the PCR-based Indel marker that can be applied at least into 20 indica rice accessions. However, to discuss the association with the phenotypic variance (i.e., photosensitivity), a more in-depth analysis such as GWAS using large numbers of varieties have been required. In contrast, because the authors investigated the indels in a small number (20) of varieties, it is difficult to conclude the association with photosensitivity. They should analyze a larger number of rice varieties to discuss the association with photosensitivity. At least, the authors should discuss the limitations of their study. Answer: Thank you for your valuable comment. We appreciate your suggestion regarding the application of Genome-Wide Association Study (GWAS) to identify candidate SNPs and Indels associated with the photoperiod-sensitive trait. However, it's important to note that our study is constrained by the limited availability of whole genome sequencing data for only 20 accessions. As a result, our current focus is primarily on filtering Indels that distinguish between the two groups.
Comment 2: In addition, the meaning of “high correlation (of the gene LOC_Os01g23780) with photosensitivity” (P7L168) is not clear. Which data and/or statistical analysis indicate the “high correlation”? The author should explain how the “high correlation” was detected more in details. Answer: We already deleted in the manuscript.
Comment 3: From a plant physiological aspect, it may be interesting that the gene LOC_Os01g23780 that would be involved in floral development in rice (Arora et al. 2007) was identified as a putative responsible regulator of photosensitivity. The authors may describe a requirement of further investigation about how the Indel in the gene LOC_Os01g23780 can alter rice photosensitivity. Answer: Thank you very much for your suggestion, we will further investigation about how the Indel in the gene LOC_Os01g23780 can alter rice photosensitivity in the next study. Minor points:
Comment 4: P2L41-42: “several well-known QTLs” → In fact, many (more than “several”) QTLs were described below (P2L42-47). The description “several well-known QTLs…” should be revised. P2L49: “Genome-wide association study (GWAS)” → genome-wide association study (GWAS) P2L52: “Single Nucleotide Polymorphism (SNP)” → single nucleotide polymorphism (SNP) Answer: Thank you very much for your suggestion, we already changed in the manuscript.
Comment 5: P4L114-115: “the https://snp-seek.irri.org/ database” → the Rice SNP-Seek Database (https://snp-seek.irri.org/) Answer: Thank you very much for your suggestion, we already changed in the manuscript.
Comment 6: Results P5L143: “focused focused” → focused P9L176: “photoperiod-sensithive” → photoperiod-sensitive P10L201: “genotypr” → “genotype” Answer: Thank you very much for your suggestion, we already changed in the manuscript.
Comment 7: Discussion: P11L226-233: A similar description about QTLs identified previously was found also in the introduction part (P2L42-47). The repetition should be avoided. Answer: We greatly appreciate your suggestion. We have already incorporated paraphrasing and additional explanations into the manuscript to avoid the repetition.
Comment 8: Tables 1-5 and Figures 1-2: More detailed descriptions about tables and figures, such as statistical analysis, data source, DNA size marker (used in Figures 1 and 2) and abbreviation, etc. should be described. Answer: Thank you very much for your suggestion, we already changed in the manuscript.
Reviewer 3 Report
Comments and Suggestions for Authors
Manuscript ID: biology-2868770-v3
Title: Identification of Insertion-Deletion (INDEL) Markers in Landrace and Improved Indica Rice (Oryza sativa L.) Cultivars
Authors: Tam and Nhan.
General comments:
The manuscript deals with the identification and analysis of indel markers in the rice land races and improved varieties, with one of the markers is projected to be useful for ‘photoperiod sensitivity’ trait screening in rice. The manuscript uses NGS-based genome sequencing of 20 genotypes and identification of Indels and presence of these in certain genes as basis of their functional importance. The current version of the manuscript is not written in a good form, suffers from several problems including, lack of rational of the study and focus of the work, repetitive content, not detailed explanation of certain analysis, less informative data content, and incoherence between different sections of the manuscript. Also, the analysis is very simple, lacks depth but some times claims a lot, which needs to be carefully addressed. Some of the key concerns are listed below and also indicate in the PDF file of the manuscript.
Section specific comments
Title: The manuscript title indicates identification of indels in two groups of rice genotypes, however it seems more focused on the indel linked to the photoperiod sensitivity. The title may need to be revised
Summary: Minor corrections suggested in summary section. The concluding statement is a very straightforward considering the manuscript content, and may need to be carefully written. Much more analysis may be needed at field level studies. It should not appear as overstated.
Abstract: Abstract looks very long and may be minimized a little bit, with proper rationale of the study in view of the known information on photoperiod sensitivity. Some key points for improvement are listed below, and also indicated in the PDF file
Lines 19-20: When rice genomes are sequenced, why the focus was on Indels instead of more sense markers like SNPs?
Lines 25-26: Is this gene (LOC_Os01g23780) not associated with any function previously, including flowering?
Line 25: The abstracts mention identification of two genes, LOC_Os01g23780 and LOC_Os01g36500, but only one has been elaborated throughout the manuscript
Lines 27-30: May be rewritten for enhanced clarity.
Line 30: What type of mapping population, and which parents were used?
Lines 32-34: The concluding statements of the abstract may be improved.
Line 35: Key words may be reassessed, as rice comes thrice. Few other relevant key words may also be added to the list.
Introduction: The length of the introduction section is appropriate, but the content may need to be improved. Some minor changes have been suggested and few are briefly mentioned here:
Lines 39-40: These may be suitable for short/long day-light period, but how these cultivars will be suitable for diverse climate is not clear? May be elaborated.
Lines 45-48: Add references in support?
Line 52: The list given in QTLs or now known as well-characterized genes.
Lines 63-71: Why low-frequency indel markers were chosen for the present study instead of high-density SNPS, is not very clear here. This must be elaborated here for the rationale of the study.
Material and methods: This section may need to be improved by providing details in several analysis. See the PDF file for comments and some points below:
Line 81: Use same designation everywhere (varieties/ genotypes/landraces/improved rice varieties).
Lines 95-96: Growth conditions may be mentioned in the parenthesis.
Line 101: Are these genome sequences submitted to any database? If yes the accession numbers/ reference numbers may be provided.
Lines 107-111: See the steps of the protocols are properly arranged, sequentially.
Lines 119-120: Kindly see the data analysis was done in 2019. It should have been updated in 2023/2024? Kindly explain?
Lines 122-127: How much confidence is there that the indels are not due to sequencing errors leading to missing reads? How it was ensured?
Lines 131-132: On what basis it was chosen that a 20bp and more long indels are appropriate for the analysis?
Lines 148-150: This methodology is not clear in M&M, some more elaboration should be included.
Results: Results will need improvement at several places. In particular, the content/description should be written in a manner to avoid unnecessary lengthy details and content which is also evident in the figure/tables. See comments/suggestions in the PDF file and also below:
Lines 153-156: Is these many indels are expected to be present among accessions of a species, such as Rice? Is any previous study also show such a high number on indels? What about already sequenced genomes of different Rice genotypes? Such High number of indels are observed?
Lines 157-158: Any reason why only four improved rice varieties are showing higher number of indels than 16 land races used.
Lines 160-161: Are a lot of genes affected on Chr1 due to so many indels?
Lines 167-168: The rationale of selecting 20 bp indels is not clear.
Lines 170-172: How it was ascertained, if an indel is insertion or deletion in a genotype/landrace?
Lines 173-175: Why S01 at serial no 2nd excluded?
Lines 182-183: The genes were associated (linked to the region/gene) with these indels or these were in the genes.
Lines 190-194: This gene (LOC_Os01g23780) is already know to be involved in flowering? How much variation (including SNPs or any other marker like STMS) are already known to be associated with this gene in previous reports? If yes, have these been used/are these useful for marked assisted selection for this trait analyzed in this study?
Line 199: Why all other indels were excluded for further analysis? How it was ascertained that these are not at all associated with the trait? This is important, as more than one loci may be crucial for the phenotype.
Lines 200-202: Check the language, it seems the 20 bp indel was located in the gene and 'not the gene located on the indel'. Kindly check and modify the description.
Lines 202-210: Description may be written in a clear and concise manner. It looks more appropriate for the M&M section.
Lines 211-212: The PCR only checks the indel region and not the phenotype diversity.
Lines 214-215: This should have been discussed in the M&M, including whether use of F1 or F2 should have been better for association studies?
Lines 227-232: The statements lack clarity, may be improved.
Lines 233-248: This is repetitive content, almost similar to previous paragraphs. It may be removed.
Lines 252-253: The rationale of this analysis is not clear? Was trait association was also observed in these accessions is not clear?
Discussion: Discussion section needs to be shortened by keeping previous literature, minimizing /removing the repetitive content, adding the references in support of important statements, and keeping the description relevant to the focus of the present work.
Lines 266-270: This is same as introduction, may be removed/minimized.
Lines 270-271: Is the known genes are there, why there is need to develop an Indel marker? Any marker, including SSRS, SNPs etc., linked to these genes can be utilized for this trait (photoperiod sensitivity) in rice. This needs to be justified.
Lines 272-281: This description is same as results section, may be removed or minimized.
Lines 285-292: This description is same as results section, may be removed or minimized.
Conclusions: Several statements in this part seems repetitive, may be removed or minimized.
Figures: There are few minor concerns related to the Figures. In general, the figure legends/captions should have been a little bit made more informative.
Figure 1: Figure 1 should indicate the following; 1: Marker labels, 2: Lane labels, and 3: instead of a big figure, regions corresponding to up to 300-400 bp may be shown
Figure 2: Figure 1 should indicate the following, 1: Marker labels, 2: Lane labels, and 3: instead of a big figure, regions corresponding to up to 300-400 bp may be shown.
Tables: The information content in the tables needs to be improved, by providing foot notes to additional information/explanation.
Table 1: Same column headings may be used in the Table1
Table 3: The column headings may be elaborated/simplified in the foot notes. All are not easy to follow for the readers.Table 4: The table may only detail the Indels associated with the genes. Other Indels may be removed as they are also listed in the previous table. How the association with the genes was ascertained? Any linkage analysis, or by proximity only? This may be elaborated. Expressed protein designation (at Sr. No 7) do not indicate any functional information. This may be removed

English language quality needs to be improved considerably.
Author Response
We have carefully revised the manuscript in accordance with the reviewers' comments.
Reviewer 4 Report
Comments and Suggestions for Authors
Please find my suggestions in the red highlight in the manuscript.
Regards,

Please use native english speaker to correct the manuscript
Regards,
Yudhistira
Author Response

(The authors gave the same response as above.)

Round 2
Reviewer 2 Report
Comments and Suggestions for Authors
The authors provided additional information in Methods, figures, and tables, contributing to improvement of their manuscript.
However, the problem of “correlation” between the gene (LOC_Os01g23780) and photoperiod sensitivity remains. To conclude that the difference (Indel) in LOC_Os01g23780 is correlated with photoperiod sensitivity, further investigations using a larger number of varieties and/or near isogenic lines (NILs) with indels of LOC_Os01g23780 should be performed. The authors had better focus on a practical benefit of the Indel markers established.
I suggest that the authors should revise the sentences which includes the word “correlation” as follows.
Abstract
P1L16: “A candidate gene on chromosome 01, featuring a 20 bp deletion in the improved rice variety group, showed a strong correlation with the flowering trait (LOC_Os01g23780)” → The gene LOC_Os01g23780, which may be involved in rice flowering, was identified in a 20 bp deletion on chromosome 01 from the improved rice variety group
Result
P7L180: “genes correlated with photoperiod sensitivity” → “candidate genes associated with photoperiod sensitivity”
P7L190-192: “Notably, one gene (LOC_Os01g23780) on chromosome 01 at the position of 13373620…13374665 exhibited correlation with photoperiod sensitivity because of the gene function about MADS-box family gene.” → Notably, one gene (LOC_Os01g23780), which encodes a putative regulator of rice flowering (Arora et al., 2007), was identified at the position of 13373620…13374665 on chromosome 01.
P9L201: “Gene LOC_Os01g23780 is high correlation with the photoperiod-sensitive, therefore…” → The MADS-box family gene LOC_Os01g23780, which may be involved in rice flowering (Arora et al., 2007), was located on the Indel region between landrace (photoperiod sensitive) and improved varieties (photoperiod insensitive), and therefore,…
P10L232: “Gene LOC_Os01g23780, highly correlated with photoperiod sensitivity, prompted the design…” → Identification of the gene LOC_Os01g23780, which might be associated with photoperiod sensitivity, prompted the design…
Author Response
Comment 1: P1L16: “A candidate gene on chromosome 01, featuring a 20 bp deletion in the improved rice variety group, showed a strong correlation with the flowering trait (LOC_Os01g23780)” → The gene LOC_Os01g23780, which may be involved in rice flowering, was identified in a 20 bp deletion on chromosome 01 from the improved rice variety group
Answer: Thank you very much for your suggestion, we already changed in the manuscript (red color).
Comment 2: P7L180: “genes correlated with photoperiod sensitivity” → “candidate genes associated with photoperiod sensitivity”
Answer: Thank you very much for your suggestion, we already changed in the manuscript (red color).
Comment 3: P7L190-192: “Notably, one gene (LOC_Os01g23780) on chromosome 01 at the position of 13373620…13374665 exhibited correlation with photoperiod sensitivity because of the gene function about MADS-box family gene.” → Notably, one gene (LOC_Os01g23780), which encodes a putative regulator of rice flowering (Arora et al., 2007), was identified at the position of 13373620…13374665 on chromosome 01.
Answer: Thank you very much for your suggestion, we already changed in the manuscript (red color).
Comment 4: P9L201: “Gene LOC_Os01g23780 is high correlation with the photoperiod-sensitive, therefore…” → The MADS-box family gene LOC_Os01g23780, which may be involved in rice flowering (Arora et al., 2007), was located on the Indel region between landrace (photoperiod sensitive) and improved varieties (photoperiod insensitive), and therefore,…
Answer: Thank you very much for your suggestion, we already changed in the manuscript (red color).
Comment 5: P10L232: “Gene LOC_Os01g23780, highly correlated with photoperiod sensitivity, prompted the design…” → Identification of the gene LOC_Os01g23780, which might be associated with photoperiod sensitivity, prompted the design…
Answer: Thank you very much for your suggestion, we already changed in the manuscript (red color).

Reviewer 3 Report
Comments and Suggestions for Authors
Manuscript ID: biology-2868770-v4
Title: Identification of Insertion-Deletion (INDEL) Markers for Photoperiod Sensitive in Rice (Oryza sativa L.)
Authors: Tam and Nhan.
General comments:
The revised version of the manuscript seems improved in writing style and description. However, the analysis is simple, lacks depth, and much more information may be needed to still use the indel to breeding applications. Some minor concerns are listed below
Section specific comments
Title: The revised title looks fine. ‘sensitive’ may be replaced with ‘sensitivity’
Figures: There are few minor concerns related to the Figures. Figure 1: Figure 1 still lack the DNA Marker size labels.
Figure 2: Figure 1 still lack the DNA Marker size labels.
Comments on the Quality of English LanguageEnglish language quality is fine, minor correction may be needed in certain sections.
Author Response
Title: The revised title looks fine. ‘sensitive’ may be replaced with ‘sensitivity’
Author's reply: Yes, it is done.
Figures: There are few minor concerns related to the Figures. Figure 1: Figure 1 still lack the DNA Marker size labels.
Author's reply: This comment is carefully addressed. We revise the text accordingly.
Figure 2: Figure 1 still lack the DNA Marker size labels.
Author's reply: Yes, it is done.
Reviewer 4 Report
Comments and Suggestions for Authors
Dear Authors,
The authors need to complement the manuscript with data on the number of days to flower each genotype in Fig 2. to prove gene of PS has been inserted in the F1
The author should have also given a brief description of the agronomy performance (number of tiller, plant high, yield) of each tested genotype it can be presented in the Supplementary file.
Any comment can be seen in the reviewed manuscript.
Regards,
Yudhis

Author Response
The authors need to complement the manuscript with data on the number of days to flower each genotype in Fig 2. to prove gene of PS has been inserted in the F1
Author's reply: Yes, it is done. The information was adding in the Supplementary Table 1
The author should have also given a brief description of the agronomy performance (number of tiller, plant high, yield) of each tested genotype it can be presented in the Supplementary file.
Author's reply: In our study, we include plant height data in the supplementary table. In our prior experiment, we omitted measurements for tiller number and yield due to the sensitivity of landrace accessions to short-day conditions, leading to variations in their agronomy depending on planting timing.